# Social Impacts of a Mega-Dam Project as Perceived by Local, Resettled and Displaced Communities: A Case Study of Merowe Dam, Sudan

Al-Noor Abdullah [1] and Sanzidur Rahman [1,2,*]

1    Plymouth Business School, University of Plymouth, Plymouth PL4 8AA, UK; alnoorabdulla18@gmail.com
2    Faculty of Economics, Shandong University of Finance and Economics, Jinan 250001, China
*    Correspondence: srahman@plymouth.ac.uk; Tel.: +44-7807654143

**Abstract:** The paper assesses social impacts of a mega-dam project (Merowe Dam in Sudan) as perceived by host and affected communities (i.e., upstream, downstream, and relocated residents), which is not commonly seen in the literature. Primary survey and interviews were conducted with 300 residents, government officials, the Dam Implementation Unit (DIU), NGOs and other key informants. Five inter-related areas of impact were scrutinized: (a) displacement of communities; (b) resettlement of displaced communities in a new location; (c) technological factors; (d) social mobility factors; and (e) economic and political institutions. Results show that Merowe Dam exerted positive as well as adverse social impacts on local communities. Increase in home sizes, opportunities for children's schooling and quality of life improvement ranked as the top three positive impacts with residents located downstream scoring relatively higher than relocated and upstream residents. Relocated residents also showed positive attitudes towards the provision of essential services (schools, health facilities, availability of running water, electricity, marketplace, etc.), thereby enabling them to enjoy and flourish in their social lives. The adverse impacts are centered on intangible factors, such as, sentimental effects closely related to their feelings, loss of history, memories, nostalgia about the old place, and grievances regarding compensation packages and its management. Therefore, it is important to recognize the need for long-term monitoring of the resettlement process and provide emotional support to those displaced and resettled. Furthermore, there is also a need to address the livelihood requirements of local communities in the affected region.

**Keywords:** displacement of communities; community perception; social impact; resettlement

## 1. Introduction

The history of dam development is linked with the evolution of its significance and impact on society and their socio-economic conditions through displacement and resettlement, especially in developing economies (Jackson and Sleigh 2000). Global economic development is experiencing changes and transitioning from the 'old school' development approach to a contemporary development method, which began in developing countries since 1980s (Amin and Thrift 1995; Behera 2006). As Shirley and Kammen (2015) suggests, mega-dams should be regarded as socio-technological systems, rooted within the socio-economic setting and co-evolving with socio-political institutions.

Therefore, mega-dam projects can have a great impact on society, both positively (Agrawala et al. 2003; Kim 2006) and negatively (Ansar et al. 2014; Flyvbjerg et al. 2003). Numerous developing countries have deployed mega-dams as a mean to generate electricity, irrigation, and water storage and to support cities and manufacturing industries. Many areas of mega-dam impacts were investigated including social impacts, environmental aversion, political corruption, governmental connections, and a philosophy of undertaking projects instigating nationalism (Everard 2013; Tischler 2013; Isaacman and Isaacman 2013; Biswas and Tortajad 2012). The social impacts of displacement, relocation, livelihood, and

changes in the socio-economic status of affected communities have also been investigated, but they do not consider how social, cultural and political conditions vary in different locations. Therefore, homogenizing the social impact of dams can undermine the many contributions dams have made for communities in different parts of the world (Varma 2003; Verhoeven 2011).

Clearly, there are many issues related to the development of dams across the world when viewed from a social perspective, and the situation in Sudan is no exception. Indeed, all dam projects in the world including Sudan's Sinnar Dam and the current Merowe Dam resulted in damage of livelihoods and displacement of entire communities (McDonald et al. 2009). Therefore, this study focuses on Sudan as one of many countries facing socio-economic stagnation. The Sudanese government utilized its limited resources to construct the Merowe Dam mainly for electricity generation to modernize Sudan's economy. This modernization is connected to numerous complementary schemes to address the Merowe region's lack of economic and social progress.

The overall aim of this study is to analyze the social impacts of mega-projects at the micro-level focusing specifically on the Merowe Dam in Sudan as the case study area. To achieve this aim, the specific objectives of this study are to: (a) identify the range of social issues related to changes the dam brought to local communities, such as displacement, resettlement, access to resources, conflicts, community fabrics, relationships, and health issues for both the displaced and resident peoples; and (b) evaluate the influence of social changes that the dam brought to the region as perceived by communities themselves. The paper provides insight into conflicts among communities and with the authorities. It further explores non-material aspects, such as culture, community health, social relationships, and social structure. The paper did not focus on the traditional approach of evaluating mega-projects, based on the economic rationality (i.e., costs and benefits analyses), rather focused on the community-level development that deals with other underlining factors related to the daily lives of the common people. The significance of micro-level assessment of this research is to avoid shortcomings of relying heavily on macro-level and massive numeric data. This approach provided details on how mega-projects influence socioeconomic development delivery at the community level, and this is the main contribution of this study to the existing literature. The perception of the community explored and verified through investigating two key areas of impact: displacement of communities and resettlement of displaced communities and other communities in the region. These two key factors exert major changes in the socio-economic conditions of displaced and relocated communities.

## 2. Literature Review

Socio-economic research regarding impacts of mega-dams largely considers the cost–benefit analysis, while studies investigating local communities' perceptions, as the key beneficiaries of these mega-dam developments, are limited. It is important to consider perceptions of the affected communities when investigating mega-dams' social and economic impacts because such approaches incorporate an evaluation of the different goals of communities, some of which have been achieved and others yet to be achieved from dam development. Displacement and relocation of communities are the main elements of mega-dams' socio-economic impact. Further impacts on communities include damages to means of support, loss of revenue, and threat to well-being (McCully 2001; Smith 1968). Therefore, as Welzel et al. (2003) suggest, livelihoods of the societies directly affected by any change in economic status quo, i.e., dam construction, affect their socio-economic conditions positively and/or negatively.

Socio-economic and human development researchers monitor these factors and their impact on communities closely. The socio-economic status of individuals and/or groups is normally determined by a collection of factors, (i.e., schooling, earnings, profession, prosperity, birth location, etc.). Mega-dam development affects these aforementioned aspects, and the direction (positive or negative) of the effect on any single element is determined by community dynamics, and the elements examined, i.e., loss of earnings, source of revenue,

etc., of the settlers and displaced communities (WCD 2000; McDonald et al. 2009; Marsh 2014). Although social and economic impacts of mega-dams were thoroughly investigated due to the surge of mega-dams after the emergence of Brazil and China as rapidly developing economies (Jackson and Sleigh 2000), evaluating perceptions of local communities only received partial consideration, despite the veracity of socio-economic elements being crucial preconditions for sustainable development of communities (Askouri 2007). This indicates organizational changes in projects occurring within international development settings, which may impact developing countries in the pursuit of their much-needed electricity for economic and social transformation (Agrawala et al. 2003; Pearce 1992).

Mega-dam studies have used cost–benefit analysis and quantitative methods as a means of analyzing the macro-economic effect of dams, using standard massive data while offering partial consideration to situations of affected communities' and their perceptions (Ansar et al. 2014; Flyvbjerg 2009; and Flyvbjerg et al. 2003). Likewise, character of the economy, phases of development, agency of society, economic development needs, stage of institutional maturity, geopolitics and socio-cultural aspects of mega dam projects have been given little consideration (Dogra 1992; Varma 1999). A few studies by researchers such as Baviskar (1999), Everard (2013), Tischler (2013), Isaacman and Isaacman (2013), and Biswas and Tortajad (2012) noted mainly negative perceptions of affected communities. This perception is due to the negative impacts of dams on socio-economic elements ranging from the loss of livelihoods to displacement of affected communities. Nevertheless, this does not reflect the complete picture of mega-dams as the extent of negative impacts differ case by case. Furthermore, mega-dams bring many positive impacts to local communities. The focus on purely negative social and economic impacts ignores numerous benefits which communities around the world have gained from mega-dams (i.e., Nepal, Turkey; Varma 1999, 2003; Lord 2016).

The discussion regarding mega-dams' social and economic impact has become increasingly polarized, particularly in recent decades. The polarization seems to mirror the frustration of people with contrasting opinions, instead of urging a useful discussion amongst stakeholders about the challenging choices regarding water, energy, and other natural resource management (Verhoeven 2011, 2015; Di Iacovo and O'Connor 2009). Hence, a generalization of social and economic impacts of dams might weaken many programs meant to support the socio-economic development of different communities around the world (Varma 2003; Kim 2006).

Governments and communities aspiring for electricity generation construct dams to improve agribusiness through institutional infrastructure and irrigation. However, this aspiration is challenged by critics, who deem this as politically motivated, because displaced communities are excluded from the decision-making processes and face injustice, especially in countries ruled by authoritarian regimes (Ahram and Goode 2016; Kleinitz and Näser 2011; Davis et al. 2009; Thomas and Adams 1999; and Dombrowsky et al. 2014).

Given this background, the present study provides a social analysis to assess contributions of mega-dams to society, thereby serving as a socio-economic development tool to improve communities through electricity generation and delivery of social infrastructure (i.e., health, education, electricity, and better living conditions), which are typically the core justification to commission mega-dam projects (Varma 1999, 2003). The socio-economic justifications are debated within the literature, where many opposing views, thoughts, claims, and counterclaims exist (Schultz 2002; Wu et al. 2013). This study aims to examine mega-dams' social impacts on the daily lives at the community level based on local people's viewpoints. Micro-level investigation in this study aims to compensate limitations of macro-level studies and reliance on massive quantitative data, where impacts of the mega-dam developments on the common people at project locations are disregarded. This research is aimed at examining socio-economic development at the community level through an in-depth analysis of how mega-dams influence the provision of such development elements. It contributes to the literature on mega-dams' socio-economic benefit sharing, future benefit, information on what went well and what went wrong, and for whom benefit

sharing schemes exist (i.e., agricultural, irrigation schemes, electricity, and infrastructure) (Varma 1999, 2003). This is a subject which is gaining scholarly interest, especially about the effectiveness and lessons learned (or not) regarding mega-dams' complex impacts.

## 3. Methodology

### 3.1. Study Area: The Merowe Region

The case study region of Merowe is located in the Northern State with a landscape consisting of hills, flat desert, and mountains. The region has a desert climate with a temperature of 46 °C in summer, which drops to 20 °C in winter (DIU 2007; Leach 1919). The area is extremely hostile, and only the lush strip of land by the bank of River Nile is habitable.

Upstream, as well as downstream, communities encountered huge impacts caused by Merowe Dam. Nonetheless, those communities and the wider nation also have acknowledged its economic and social benefits (Varma 2003; Shirley and Kammen 2015). The Merowe region was inaccessible before the dam, and people mainly resided in small communities by the bank of the River Nile and the small islands (Barbour 1966). Between 2006 and 2009, the Merowe Dam and its reservoir affected approximately 30,000–50,000 households from Hamdab Amri and Manasir communities causing, dislocation, loss of historical artefacts, date-palm trees, source of income, and damage to livelihoods. Likewise, assessing how the new and old settlements were impacted in terms of farming is central for the affected communities (Bosshard 2007). Dam research often focuses on upstream displaced communities and other communities around the reservoir (De Jalon et al. 1994; Lessard and Hayes 2003). This study extends its coverage to include communities downstream (Hamdab West, Al Degawit and Nouri) to investigate the impact of the new water regime on farming and the area of cultivated land.

The variation of water flow frequently affects downstream communities and may impact fishing communities, the ecosystem, and farm production (De Jalon et al. 1994). This emphasis requires consideration of environmental as well as social and economic impacts (Power et al. 1996). These communities assisted the researcher to catalogue and evaluate Merowe Dam's socio-economic impact on their lives. According to Welzel et al. (2003), even marginal alterations in economic conditions for downstream and upstream communities can have enormous livelihood repercussions and accordingly, on their socio-economic status and human development. Merowe region is considered as an administrative unit, economically and socially, making it a suitable case to study the changes that occurred during the Merowe Dam's lifecycle (Rigg et al. 2012).

### 3.2. Research Method: The Case Study Approach

Case study is a vital method in social research and Yin (2003) describes it as 'an empirical inquiry that investigates a contemporary phenomenon within its real-life context, especially when the boundaries between phenomenon and context are not evident' (p. 35). Other academics believe that case studies can be applied to all research categories and types (Yin 2003; David and Sutto 2004). Yin (2003) considered a case study to be an all-encompassing method—covering most research elements from designing to precise strategy for data analysis (p. 35). The case study of Merowe Dam is selected due to its representation and becoming a meeting point for economic, social, and government forces, which led to environment and change in culture (Stake 2000; Rigg 2007). A further motive of selection is Aswan High Dam's historical impact in the region, and the possible rekindling of grievances, especially with more dam projects planned, i.e., Kajbar dam (see Figure 1).

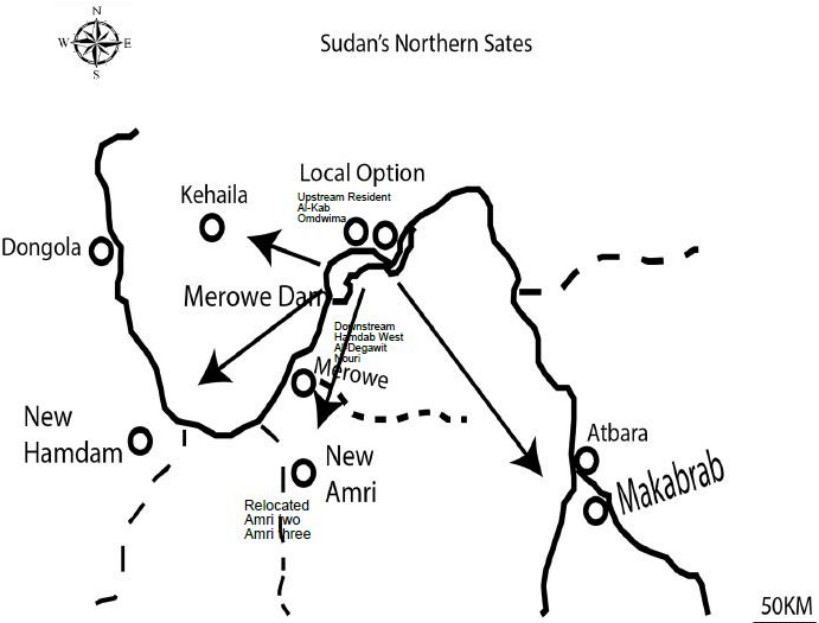

**Figure 1.** The study locations, downstream and upstream. Source: Author.

Unexpected shifts in government strategies in international investment for mega-projects and their social and environmental impacts, ignite regional and global debate (i.e., ideological, economic, and political). The Merowe Dam construction started in 2004 and became operational in 2009, becoming one of the largest dams in the Nile (Askouri 2007). It ignited economic and political struggles between downstream and upstream countries—Ethiopia and Egypt—especially in the context of the Grand Ethiopian Renaissance Dam construction.

### 3.3. Data Collection

Data collection was conducted by the main author from April to September 2017 in the Merowe region, exploring the effect of Merowe Dam on various localities in three districts from Sudan's northern states—upstream, downstream, and upstream-relocated residents (see Figure 1). The study used qualitative and quantitative methods. Over 100 settlements and islands were affected by Merowe Dam. The government created four settlements to relocate most of the affected communities. Nonetheless, several affected people rejected resettlement, favoring reservoir periphery as their home. A representative sample of four villages of displaced communities were selected, including resettled villages and those who rejected resettlement. Three downstream settlements were selected as the impact on them is less since no displacement occurred, in order to draw a comparison amongst the three different locations. In total, seven settlements were targeted to collect samples, three settlements from downstream communities (namely, Al-Degawit, Nouri, and Hamdab West), two upstream-residents by the reservoir area (namely, Al-Kab and Omdwima), and finally two upstream-relocated villages (namely, Amri-2 and Amri-3). The total sample frame was 30,000 households of which 300 families were selected. The multistage sampling strategy was used for data collection including clustering samples randomly (100 per district distributed amongst settlements equally). The seven settlements were assembled into four unique clusters based on variations in socio-economic and terrain characteristics. An equation was used to select the 300 samples (Equation (1)) as suggested by (Ahmed et al. 2013). The sample suggested by the calculation was 264, which however, increased to 300 to cover possible missing samples.

$$n = \frac{Nz^2 p(1-p)}{Nd^2 + z^2 p(1-p)} \tag{1}$$

where:

*n* = sample size
*N* = total number of households (1000 per community)
*z* = confidence level (at 95% level *z* = 1.96)
*p* = estimated population proportion (0.5, this maximizes the sample size)
*d* = error limit of 6% (0.06)

A thorough survey of inquiry was implemented in the study areas, and to eliminate any social or geographical predisposition, a cluster sampling method was applied. Fifty households were randomly selected from each of the four settlements from the two upstream areas. For the downstream location, 33 households were randomly chosen from two settlements and 34 households from one settlement.

The questionnaires were deployed based on the location characteristics, the degree, and type of impacts. The relocated communities are New Amri 2 and Amri 3, and the upstream communities of the reservoir areas are Al-Kab and Omdwima, who did not relocate, and the communities from downstream area of the Nile are Hamdab West, Al-Degawit, and Nouri (see Appendix A). These communities differ in characteristics and the degree of impact that the dam exerted on them. This variability enabled the research to contrast experiences of downstream and upstream communities of the Merowe Dam. The researcher and four research assistants administered the questionnaires over a period of two months. Questionnaires were delivered to every second household in the seven selected communities, depending on the community size. This system was used to avoid any bias and to ensure that the selection was random and represent diversity of the participants. The questionnaires were mainly administered during afternoon and evening hours in order to ensure that as many family members were available to proceed with the questionnaire or to arrange a second visit when family members would be available.

The survey questionnaire collected information focusing on communities' socio-economic characteristics, property, possessions owned in addition to their attitudes regarding Merowe Dam and its effect on homes, health, relationships, livelihood, etc. Moreover, a cluster of 18 factors (displacement and resettlement related) associated with Merowe Dam's social effects was established. The survey contained a series of socio-economic indicators identified from the literature review, and other indicators specific to the Sudanese context, to allow respondents to express their opinions on Merowe Dam (Loney 1995; Becker and Murphy 2009). The impact of Merowe Dam on socio-economic conditions is scaled on five points ranging from 1 to 5: where a score of 1 represents strongly negative and a score of 5 represents a strongly positive impact. This qualitative data collection technique is appropriate as it is simple to understand and to draw conclusions following results from the responses. The participant's perceptions of the socio-economic impact were decided after a thorough analysis of a total of 18 socio-economic indicators associated with displacement and resettlement (see Tables 1–3).

**Table 1.** Summary statistic of the demographic characteristics in three districts.

| Items | Unit of Measurement | Downstream | Relocated | Upstream | Total |
|---|---|---|---|---|---|
| Age of head | Years | 40.79 (11.34) | 36.85 (11.83) | 34.89 (10.66) | 37.51 (11.52) |
| Education of head | Completed years of schooling | 13.15 (3.36) | 13.56 (3.31) | 14.05 (3.19) | 13.59 (3.30) |
| Male respondent | Percentage | 64% | 73% | 74% | 70.3% |
| Female respondent | Percentage | 36% | 27% | 26% | 29.7% |
| Date-palm farming | Percentage | 77% | 67% | 77% | 73% |
| Vegetable farming | Percentage | 62% | 60% | 48% | 57% |
| Arable crop farming | Percentage | 79% | 69% | 70% | 72% |
| Livestock farming | Percentage | 40% | 26% | 24% | 30% |
| Number of samples | | 100 | 100 | 100 | 300 |

Note: Figures in parenthesis are standard deviation. Source: Author's calculation.

**Table 2.** Participants' perception of Merowe Dam's social impact by locations.

| Location | | Feelings about Merowe Dam | | | Waterborne Diseases | | Social Conflicts | | Relationship between Communities | | | Total |
|---|---|---|---|---|---|---|---|---|---|---|---|---|
| | | Very Much Like | Like | Dislike | Yes | No | Yes | No | Not Good | Good | Very Good | |
| Downstream | Count | 26 | 49 | 25 | 70 | 30 | 34 | 66 | 20 | 38 | 42 | 100 |
| Upstream-Resident | Count | 12 | 24 | 64 | 61 | 39 | 57 | 43 | 28 | 30 | 42 | 100 |
| Upstream-Relocated | Count | 33 | 24 | 43 | 67 | 33 | 53 | 47 | 8 | 30 | 62 | 100 |
| Total | Count | 71 | 97 | 132 | 198 | 102 | 144 | 156 | 56 | 98 | 146 | 300 |
| | Chi-Square | 39.867 *** | | | 1.87 ns | | 12.10 *** | | 17.643 *** | | | |

Note: *** = significant at 1% level ($p < 0.01$); ns = not significant. Source: Author's calculation.

**Table 3.** Participants' perception of Merowe Dam's social impact by gender and education level.

| | Items | Feeling about Merowe Dam | | | Waterborne Diseases | | Social Conflict | | Relationship between Communities | | | Total |
|---|---|---|---|---|---|---|---|---|---|---|---|---|
| | | Very Much Like | Like | Dislike | Yes | No | Yes | NO | Not Good | Good | Very Good | |
| Gender | Male | 47 | 66 | 98 | 134 | 77 | 121 | 90 | 15 | 55 | 141 | 211 |
| | Female | 24 | 31 | 34 | 60 | 29 | 50 | 39 | 6 | 35 | 48 | 89 |
| | Total | 71 | 97 | 132 | 194 | 106 | 171 | 129 | 21 | 90 | 189 | 300 |
| | Chi-Square | 0.408 ns | | | 0.518 ns | | 0.852 ns | | 0.070 *** | | | |
| Education Level | Primary | 8 | 8 | 18 | 24 | 10 | 21 | 13 | 2 | 6 | 26 | 34 |
| | High | 63 | 89 | 114 | 170 | 96 | 150 | 116 | 19 | 84 | 163 | 266 |
| | Total | 71 | 97 | 132 | 194 | 106 | 171 | 129 | 21 | 90 | 189 | 300 |
| | Chi-Square | 0.446 ns | | | 0.443 ns | | 0.551 ns | | 0.210 ns | | | |

Note: *** = significant at 1% level ($p < 0.01$); ns = not significant. Source: Author's calculation.

Moreover, rigorous interviews were conducted using the semi-structured interview technique with 33 key actors in Khartoum and Merowe region with different levels of involvement with the Merowe Dam (Appendix B). These are: 15 university scholars, members of civil society and NGOs, 9 government representatives (ministers, policymakers, and Merowe Dam authorities), 6 veteran farmers and leaders of settlements, and 2 entrepreneurs. This selection allowed the space to deeply explore expressions (i.e., feelings,

thoughts) of the interviewees enabling the researcher to triangulate survey data results (David and Sutto 2004). Nevertheless, interview techniques (semi-structured interviews) have disadvantages in that they can be complex, suffer from predisposition, and interviewees may respond just to satisfy the interviewer (David and Sutto 2004). An example of a completed transcript from in-depth interview is presented in the Supplementary Material.

## 4. Results

Prior to presenting results of the main survey, some basic demographic features of the respondents of the study area are presented.

### 4.1. The Demographic Characteristics of the Study Areas

The basic demographic information of the respondents presented in Table 1 suggests that the mean age is 37.5 years and mean educational level is 13.6 years, suggesting high level of university attendance among participants (scores: university = 16; high school = 12, and secondary = 9). Since majority of the participants are educated, they may be well informed about Merowe Dam's impact. The gender distribution of the participants indicates the dominance of males, a reflection of Northern Sudan's tradition and culture where women's participation in political and public affairs is limited. In addition to the cultural factors, participation of women in this research may be due to the influence of the head of household and his political view and social status. Consequently, the researcher has to interview women in the presence of the head of household, which suggests some level of influence by the family views on the matter discussed. Furthermore, Table 1 suggests that agriculture is the predominant economic activity with 70% engaged in farming, especially date-palm trees, which is of economic and social pride for Northern Sudanese communities.

### 4.2. Perceptions on the Social Impacts of Merowe Dam

The conceptualization of mega-dams' social impacts is inherently complex, with a multitude of socio-economic effects occurring over time, space, tradition, belief, and value dimensions (Ersumer 1999; McCully 2001; Varma 1999, 2003). The analysis of Merowe Dam revealed that mega-dams produce both negative and positive effects for societies as they deal with complex socio-economic issues, consequently, generating disparities between more fortunate beneficiaries (downstream, upstream-relocated, and the country) and some unfortunate beneficiaries (displaced and upstream residents). However, the variation in socio-economic benefits between the two groups is negligible when the numbers of vital social projects created in the area were considered

The data suggest that people across three districts acknowledge valuable social and socio-economic improvement opportunities that Merowe Dam and its accompanying projects provided to local communities. This view corresponds with Varma's (2003) views on dams' social benefits to hosting communities. Table 2 showed diverse feelings about Merowe Dam among locals, the region, and the country. Local participants' responses reveal different perceptions within districts; however, most participants appreciate the dam despite displacement grievances and its impact on their socio-economic status. The diverse perception reflects a diversity of views within the literature, but this result to some extent contradicts largely negative views on dams, such as, by Miescher and Tsikata (2009), Loney (1995) and Flyvbjerg (2009). The participant's perceptions were generated through an array of questions answered by the participants (see Appendix A). The variation of perceptions on different questions concerning Merowe Dam's contribution is significantly different across respondent's localities (Table 2), regardless of their level of education and/or gender. The results of Table 3 show that cross-analysis by level of education and gender of respondents for the same questions show insignificant differences in perceptions. In other words, regarding Merowe Dam's contribution, perceptions vary significantly based on respondent's location, but people living in the same settlement possess the same perception regardless of their levels of education or gender. This two-level analysis enhances the credibility of the findings based on the three distinctive localities, i.e., upstream resident by

the reservoir, upstream-relocated, and downstream, capable of picking up the correct type of responses.

This reinforces the argument that local communities are not opposed to Merowe Dam itself but largely unsatisfied with how the government managed the displacement and compensation process (Table 2). The displacement and its effects on communities socially and economically combined with a lack of will from the government to address its underlying issues drove negative sentiments, especially within Manasir communities. There was also evidence of a strong association between displacement resentments and social conflicts, especially between the government and Manasir communities and within Manasir communities themselves. This is in line with Jia et al. (2011) research on dams causing social conflicts. The analysis showed significant differences between districts in the level of conflict and its impact on the relationships between members of families and communities (see Table 2). Upstream-residents (Manasir) and some upstream-relocated (Amri) communities were widely affected by the levels of conflict unlike downstream communities, but they still support their counterparts in other districts.

Concerning environmental impacts of Merowe Dam, the analysis revealed that its ecological impact seems to be more likely balanced in both downstream and upstream. Nonetheless, the topography and climate of the Merowe Dam's location played an important role in reducing its environmental impacts in the region. Furthermore, the analysis suggests that the Merowe Dam has both positive and negative ecological influences in downstream as well as in upstream locations. Nonetheless, both locations have benefited from huge fishing wealth, less erosion, and improvement in the whole ecosystem through water availability. This suggests that having a stable environment and diverse ecological system (including a supply of water for fish farming) with a stable political system could improve agricultural economy in the region. However, pertaining to health hazards, results show that the dam indirectly caused some illness related to the underground well used for drinking water and an increase in waste and flies.

Displacement and resettlement caused by the dam's construction is a common phenomenon across the world, regardless of displacement and its nature. However, socially, there are differences in resettlement contexts and the degree of satisfaction with resettlement in providing better living standards and conditions, thereby allowing displaced communities to restore many aspects of their lives.

## 5. Displacement Impact

The displacement impacts of large dams mainly divided into two groups: the effect on material possessions and property, which can be mitigated by compensation, good preparation, and implementation; and the effect on sentimental possessions, which is almost impossible to replace or mitigate (Isaacman and Isaacman 2013). This paper analyzed displacement through different aspects of the dam and their implications, which touched the lives of the displaced and surrounding communities in many respects and changed their social lives including health, housing structure, culture, and identity.

The Merowe Dam displacement began when the floodgate closed in July 2008 and the water rose rapidly. Those residing in settlements alongside the River Nile, were caught unprepared for the sudden flood. Those who were slightly prepared escaped with limited belongings and made makeshift shelters far from the river. Eventually, many of the displaced settled with relatives who previously moved to resettlement villages established by the Sudanese Government, in remote locations in the desert, far from the river (Interviewee 20, see Figure 1). There are three groups of displaced: the Hamdab communities previously living at the center of the dam's location were relocated to Al-Multaga; the residents of Amri living nearby the dam's location were relocated to New Amris; the inhabitants of the lake are Manasir communities. About 60% of Manasir refused to migrate, demanding to stay near the lake because of the estimated 15,000 tons of fish and 60,000 acres of fertile land good for agriculture. However, the other 40% relocated to Al-Makabrab and Kehila, where the compensation land at upper terraces are less fertile,

and the high cost of irrigation makes it difficult for the displaced to fully restore their livelihood.

Resentment due to dam displacement is a global phenomenon and underpinned by anger (Singh 1997). No dam is welcome by the displaced people. To explore communities' perception and sentiment on displacement issues, questions on how people feel about the presence of Merowe Dam were asked. Chi-square analysis was performed to examine any relationship connecting communities' apparent feelings towards Merowe Dam across locations, which demonstrated a significant difference ($p < 0.10$–$p < 0.01$) between locations on the feelings by both displaced and non-displaced communities.

Table 4 demonstrates negative feelings about Merowe Dam from upstream-residents, with 64% feeling *'dislike'* towards the dam. While upstream-relocated participants show some lenience toward positive feelings, especially when combining both *'strongly like'* and *'like'* 57%. However, the aggregate result of *'strongly like'* and *'like'* indicates the positive feeling (55.6%) compared to *'dislike'* at 44.4%. This research suggests displacement and a lack of readiness to tackle the underlying issues sufficiently support the negative sentiment about Merowe Dam, particularly among Manasir communities at the upstream-residents where disputes about resettlement and compensation are still ongoing. The result reinforces the views of those who believe that dams for good reasons are unwelcome by local communities such as Baviskar (1999), Everard (2013), Tischler (2013), Isaacman and Isaacman (2013), and Biswas and Tortajad (2012). Many participants and interviewees, especially from Manasir communities, expressed their disappointment about the displacement despite their high anticipation and knowing that displacement is imminent. A local farmer responded:

**Table 4.** How do you feel about the presence of Merowe Dam?

| Location | | Strongly Like | Like | Dislike | Total |
|---|---|---|---|---|---|
| Downstream | Count | 26 | 49 | 25 | 100 |
| Upstream-Resident | Count | 12 | 24 | 64 | 100 |
| Upstream-Relocated | Count | 33 | 24 | 43 | 100 |
| Total | Count | 71 | 97 | 132 | 300 |
| | Chi-Square | | 39.867 *** | | |

Note: *** = significant at 1% level ($p < 0.01$). Source: Author.

> *"The Merowe Dam is a failure and the government should review its policies toward displaced people or it will not be able to conduct any future projects".*

However, other respondents, especially from Amir and Hamdab communities, have a more moderate reaction to displacement. Another local farmer stated:

> *"The Merowe Dam is a success and provides development means for the region, but it was bad for the displaced people because we have been treated unfairly and oppressed by DIU".*

This demonstrates the degree to which people in the region view Merowe Dam as a means for socio-economic development but is also an indication of the authorities' mistreatment and unfulfilled promises made to displaced people, which have a greater influence on the way these communities perceive the impact of the dam. Furthermore, this challenges the argument that Merowe Dam was unsuccessful and confirms its success as a development project instead. Varma (1999, 2003) argues that local politicians, NGOs, and intermediaries can confuse and exploit dam-affected people for their own benefits by criticizing dams for causing displacement while doing little to support displaced people themselves. To further explore how the displaced perceive the impact of Merowe Dam on their social lives, Table 4 shows the ranking of the mean values for individual indicators on the Likert scale of 1–5 (1 = strongly disagree and 5 = strongly agree), (see Q12 of the questionnaire in the Appendix A). Analysis of community coherence, social status, farmland as a measure of wealth and pride, archaeological and historical sites, feeling of belonging to the place, and

self-esteem showed a high level of differences based on the Kruskal–Wallis tests performed across the three locations for each indicator ($p < 0.01$). Table 4 shows a significant level of agreement for downstream and upstream-relocated participants, with a moderate level within upstream-resident on positive social influences of Merowe Dam.

The mean scores show downstream participants hold a positive sentiment towards all social indicators (score ranging from 2.88 to 4.02), which can be explained as a result of not being affected by displacement and gaining more through social development projects. However, upstream-relocated and upstream residents where displacement took place, scored 1.90–3.29, indicating some resentment of the social effects of displacement. The overall mean score was very positive on social matters at the three locations (2.37–3.30, with a standard deviation of 1–1.60), meaning the responses are closer to the mean. The overall ranking of the indicator in Table 5 shows that community coherence is at the top and self-esteem at the bottom. Despite conflict associated with the dam, this result is an indication of local communities' ability to reconcile themselves. Archaeological and historical sites, feelings of belonging, farmland as a source of pride, and self-esteem, all scored slightly low (see Table 5).

**Table 5.** Ranking of social indicators impacted by displacement by districts.

| Sl. No. | Social Impact of Merowe Dam Social Indicators | Index Weighted by the Rank of Responses | | | | | | | | Kruskal Wallis ($\chi^2$) |
| | | Downstream | | Upstream-Resident | | Upstream-Relocated | | All Region | | |
| | | Index | Rank | Index | Rank | Index | Rank | Index | Rank | |
| 1 | Community coherence | 3.98 | 2 | 2.63 | 2 | 3.29 | 1 | 3.30 | 1 | 42.892 *** |
| 2 | Social status | 4.02 | 1 | 2.51 | 4 | 3.09 | 3 | 3.21 | 2 | 61.834 *** |
| 3 | Land as wealth/pride | 3.53 | 4 | 2.19 | 5 | 2.70 | 4 | 3.13 | 3 | 47.411 *** |
| 4 | Historical site | 2.88 | 6 | 1.92 | 6 | 2.32 | 6 | 2.85 | 4 | 33.881 *** |
| 5 | Feeling of belonging | 3.24 | 5 | 2.83 | 1 | 2.49 | 5 | 2.80 | 5 | 13.876 ** |
| 6 | Self-esteem | 3.56 | 3 | 2.56 | 3 | 3.26 | 2 | 2.37 | 6 | 25.249 *** |

Note: *** = significant at 1% level ($p < 0.01$); ** = significant at 5% level ($p < 0.05$). Source: Author's calculation.

This shows a clear link with Scudder (2012), Bakken et al. (2013), and Barnaby Dye (2019), on sociocultural impacts of Merowe Dam on local farming communities. These indicators are very important in northern Sudanese culture due to a long history of association with the Nile, agriculture, and land, all viewed as sources of wealth and pride. It is not surprising that archaeology and self-esteem scored lower because of their interconnection with other indicators. Most participants and interviewees expressed their concerns over these indicators. For example, a fear of change or the unknown and mixing with other tribes led to relocation resistance within Manasir communities.

Analysis of the exact ranking of Merowe Dam's social impact was conducted on displacement indicators classified by level of education and gender, where the result suggests insignificant ranking variances. This suggests that the effect of Merowe Dam is perceived differently according to location, and communities living within the same location experience similar impact regardless of the level of education and gender.

Spearman rank correlation analysis of social indicators was conducted to detect if community perceptions regarding these effects are consistent and related. The result in Table 6 demonstrates a strong positive correlation amongst all seven indicators. This result provides overwhelming confidence in the test, suggesting reliability and robustness in the ranking of the perception of communities across all three locations; downstream, upstream resident, and upstream relocated.

**Table 6.** Correlation among social indicators impacted by displacement by districts.

|  | Community Coherence | Social Status | Farmland as Wealth and Pride | Archaeological and Historical | Feeling of Belonging | Self-Esteem |
|---|---|---|---|---|---|---|
| Community Coherence | 1.000 |  |  |  |  |  |
| Social Status | 0.771 | 1.000 |  |  |  |  |
|  | 0.000 | 0.000 |  |  |  |  |
| Farmland as wealth and pride | 0.592 | 0.712 | 1.000 |  |  |  |
|  | 0.000 | 0.000 | 0.000 |  |  |  |
| Archaeological and historical | 0.399 | 0.465 | 0.556 | 1.000 |  |  |
|  | 0.000 | 0.000 | 0.000 | 0.000 |  |  |
| Feeling of belonging | 0.293 | 0.294 | 0.344 | 0.346 | 1.000 | . |
|  | 0.000 | 0.000 | 0.000 | 0.000 | 0.000 |  |
| Self-esteem | 0.546 | 0.596 | 0.532 | 0.263 | 0.275 | 1.000 |
|  | 0.000 | 0.000 | 0.000 | 0.000 | 0.000 | 0.000 |

Note: The second row shows *p*-values ($p < 0.001$). Source: Author's calculation.

The results of Tables 4 and 6 suggested sentiments regarding six social indicators are very strong amongst upstream residents and upstream-relocated participants concerning changes that might come with displacement. A local farmer respondent states:

*"As Manasir, we are not against the dam, but we want to stay on the land and at the place, we belong to and we don't like to mix with other tribes." "Amrian culture was affected by displacement a great deal; our lives depend on farming. After displacement, the compensated land has low quality with insufficient irrigation water supply. I lost 40% of 100 date palm trees and my income reduced by about 50%, consequently I rely on other sources for income".*

However, many participants realize the opportunities that displacement provides through relocating and mixing with other communities: enriching their culture and gaining new life experiences. As local farmer interviewee downplayed the fear of mixing with other tribes:

*"I believe we miss our land where we were born but, displacement has some social benefits. Relocating allowed us to mix with other tribes because the dam brought many Nile-based communities together in one place. We managed to marry from other communities such as the Robatab and learning new cultures and traditions, which we are not accustomed to".*

Other participants agreed with the criticism of dams suggesting that DIU did not pay attention to the needs of people in their displacement methods, and their focus was how to evacuate people and get the place ready for building the dam. A local committee member interviewee elaborates:

*"We are not opposing the displacement, but we requested it is conducted on mutual terms with enough time to prepare, but we realise that DIU only cares about building the dam at any cost. To expel us from our homes, they closed the dam and submerged 2800 families without warning".*

However, DIU official interviewee rejects this claim and states:

*"Our main concern is the resettlement of the population and fully supporting them to settle into their new location, and then we give permission to start building and subsequently filling the lake".*

In contrast to the official view, most observers and experts believe that the government and its DIU wing in charge of water management projects treated the displaced as subjects

without negotiation or proper consultation rights. Even those who were consulted felt their opinions did not carry much weight. A university academic interviewee suggests:

> *"At the start, everyone in the region welcomed the dam and there was no resistance or opposition to displacement until further down when the citizens, especially Manasir communities, realised how the DIU treated them with no respect and as subjects to be brought to modernity kicking and screaming".*

Displacement is a highly controversial subject in dam studies on how it impacts materialistic and sentimental social aspects of displaced and local communities. In respect to Merowe Dam, local committee member interviewee agreed with the literature such as Verhoeven (2011), Loney (1995), and Kleinitz and Näser (2011). He states:

> *"Naturally, people do not forget their birthplace. I lived near the Nile, I heard water sound, and I can drink from the Nile with my bare hands. Every morning I would go and sit by the Nile; I remember the places of mountains and the burial places. I miss it because of the displacement, but now our lives are better in many aspects compared to the past, and the rest is memories."*

Displacement and resettlement processes have caused conflicts between Manasir communities and the DIU and within communities. There are claims and counterclaims of who caused the conflict but underpinning them are the communities' dissatisfaction with the way DUI dealt with displacement. Many participants claimed that they received no support from the government or NGOs because of government restrictions in place to enforce the evacuation. However, many government officials, locals, and experts interviewed referred to non-financial types of support, which were provided to displaced people, including emotional support, comfort, food provision, etc. To have a better understanding of social issues related to Merowe Dam, it is essential to study displacement in association with resettlement, which is elaborated below.

## 6. Resettlement Issues

The growing desire for more energy sources drives voluntary and involuntary resettlement and is becoming a massive dilemma that developing economies face (Jaffee 1998). Governments pursue modernization and socio-economic development, but they must pay a substantial social price for dam development. The price of such projects is tremendous, yet social factors always receive less consideration (Scudder 2012). Governments around the world, international organizations such as the World Bank and NGOs have acknowledged the economic and social effects of mega-dams in the form of voluntary and involuntary resettlement, which are by no means the only major side effects of dam development (Murdoch 1994). Dam resettlement is a necessary condition of socio-economic development, but the consequences differ from one country to another due to the scale of the projects, magnitude of displacement, local conditions, and dam specification (Murdoch 1994). This research finds that the international settlement debate and matters seeking standardization of resettlement are more harmful than beneficial to displaced people. In Merowe, the living conditions of displaced people is much better post-Merowe Dam in many aspects: socially and economically, provision of services, and living conditions. However, this by no means undermines the agony of property lost and the disruption of social life the displaced communities have experienced, especially the emotional distress, which cannot be compensated. The magnitude of mega-dam criticism partially feeds into residual resettlement difficulties, especially for Manasir communities. Therefore, more focused discussion among researchers and practitioners on localization and contextualization of resettlement is needed now more than ever. Without this discussion, the debate, despite good intentions of both sides, does little to support the predicament of the millions affected by mega-dams.

The Merowe Dam resettlement villages were constructed within two different states and in four areas, as presented in Figure 1. Hamdab communities made up about 7% of the total displaced people who lived at the dam location and were relocated to "New

Hamdab" or "Al-Multaga in 2003 in the Northern State, roughly 45 km away from the dam. A few people in these communities refused resettlement and favored settling at the reservoir shoreline or moved in with relatives in the region. Around 2800 displaced Amri families (accounting for 28%) were displaced following the unexpected flooding in 2006 when the floodgates were closed temporarily. They mostly resettled in the four villages of "New Amri Villages", (No. 1–4) many kilometers away deep into Bayouda desert within the Northern State. About 15% rejected the resettlement (local committee interviewee, see Figure 1). The Manasir communities who lived further upstream from the dam were displaced when the floodgates were closed following the dam's completion in late 2008. Around 5000 families from Manasir villages have been flooded or partly flooded. Manasir communities account for 65% of the total displaced population (Figure 1).

The Manasir displaced people were offered resettlement in six new villages in the River Nile State (DIU 2007; Askouri 2007), but two-thirds rejected the resettlement. Some migrated to Khartoum, but the majority still live-in new villages established by the displaced along the reservoir coast, or on higher ground of semi-submerged area near their original settlements known as "Alkhiar Al Maghali" (meaning local choice) defined here as 'upstream-resident'. These communities managed to live on fishing and agriculture from the fertile floodplain land. The 'upstream resident' communities are eligible for full compensation for property lost. Only a few received monetary compensation for date-palm trees, with no broader compensation package, such as new homes, services, etc. However, some sections of the community (nicknamed "Karazies", a degrading term meaning traitor) have accepted the compensation package and resettled in Makabrab and Kehila. This caused a strain in social relations with their neighboring communities upstream. The government has a favorable perspective of its resettlement package as the most adequate ever created. A DIU official states:

> "The selection of the locations and house design was decided in consultation with displaced people according to the design of existing housing and materials in the region. The components, interior design, furniture, and space are expanded and updated to modern national standards. We have also built excellent public services, such as schools, hospitals, social and public services like mosques, young people's culture centres, women's centres and nurseries".

In addition to the government's claims of social, economic, environmental, and political success of the Merowe Dam project, it also claimed that lessons were learned from the Aswan High Dam experience by resettling displaced people within the same environment, culture, and region. However, the government's disputes with the Manasir concerning compensation packages and resettlement location cast doubt on the government's claims of success. The majority of displaced Manasir's accepted parts of the compensation package, except the resettlement location, which is at the center of their dispute with the government. The displaced suggested resettlement by the reservoir where they live currently, but the government did not agree to this because of technical reasons, which considered dubious by the settlers. As the government continues to refuse the Manasirs' demands to resettle by the reservoir, the chance of finding a solution becomes slimmer.

Furthermore, the DIU not only refused to negotiate with upstream residents, but it also tried to disrupt all attempts made by mediation committees to resolve the issue. This not only contradicts the government's claim of participation and bottom-up development approach, but it also suggests that development in non-democratic states is mainly dictated by the state and citizens have little or no input (see Figure 2). This is supported by the literature, as Scudder suggests, resettlement stresses are a form of deprivation by removing economic, social, and cultural resources as well as political power and, most profoundly, the determination to decide where and how to live. In fact, as many local participants, expert interviewees, and observers agree, some success has been achieved concerning the settlement. Those who disagree with this refer to the Manasir issue. A local farmer and a local committee member interviewee state:

*"It is human nature not to forget one's birthplace and I am emotionally affected by the displacement. I do miss my old place but, when I compare our conditions in the past to now, the new settlement is better economically, socially and services are available. My feeling is that everyone who migrates from a village to a city will not forget where they came from, but life is very easy at the new settlement".*

*"In the past, we lived traditional and basic lives. We were closed societies resisting change. Now our attitude has changed: we are open to modernity, education, and technology, and 99% of communities own fridges, electric cookers, TV receivers and washing machines. All these are available to us, and life is easier. The communication companies have created mobile networks, which we did not have at the old settlement"*

These opinions about the new settlement are supported by many respondents who challenge the critics of dam resettlement, such as Scudder (2012) and Jackson and Sleigh (2000). Singh's study of 50 large dam resettlements in five continents (1997) acknowledged that living standards in resettlement areas have improved slightly and livelihoods have been restored to some degree. The Sudanese government's misrepresentation of the Merowe Dam resettlement through its control of the media portraying the substantial facilities of the new houses, land, services, and financial compensation misled the Sudanese public and encouraged a dialogue that was critical of the Manasir, and others affected who protest or show dissatisfaction with resettlement.

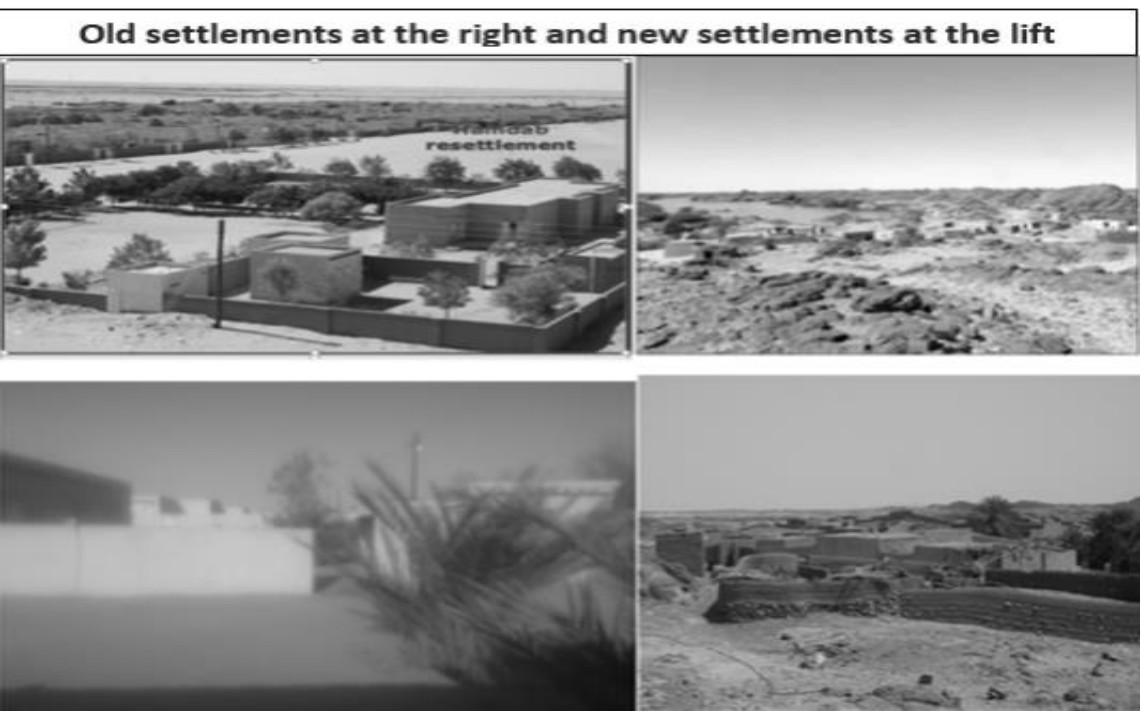

**Figure 2.** Images of new settlement and old settlement: Source: Author.

An analysis of participants' responses provides a clear picture of settler's perceptions of the Merowe Dam resettlement using a ranking of Likert scale of 1–5 (1 = strongly disagree and 5 = strongly agree), of home size, schools, and quality of life. The indicators show a significant level of agreement among downstream respondents that Merowe Dam had a positive influence on these social indicators. However, upstream-relocated and upstream residents show a moderate level of agreement on the positive social influence of Merowe Dam. The test shows a significant difference between the locations ($p < 0.001$) for all three indicators (Table 7). The mean comparison test of home size, schools, and quality of life provide important insights into participants' opinion. As expected, mean values of downstream location are higher (3.64–3.89) for all three indicators, especially the quality of

life and schooling. This is due to improvements in health, education infrastructure, and income, helping people to have access to modern building materials in the downstream district.

**Table 7.** Ranking of resettlement living suitability indicators by districts.

| Sl. No. | Social Impact of Merowe Dam Social Indicators | Index Weighted by the Rank of Responses | | | | | | | | Kruskal Wallis ($\chi^2$) |
|---|---|---|---|---|---|---|---|---|---|---|
| | | Downstream | | Upstream-Resident | | Upstream-Relocated | | All Region | | |
| | | Index | Rank | Index | Rank | Index | Rank | Index | Rank | |
| 1 | Home sizes | 3.640 | 3 | 2.230 | 2 | 2.550 | 2 | 3.160 | 1 | 55.81 *** |
| 2 | Children school | 3.840 | 2 | 2460 | 1 | 3.180 | 1 | 2.807 | 2 | 43.40 *** |
| 3 | Quality of life | 3.890 | 1 | 2.040 | 3 | 2.440 | 3 | 2.790 | 3 | 86.50 *** |

Note: *** = significant at 1% level ($p < 0.01$). Source: Author's calculation.

Upstream residents scored lower (2.00–2.46) for these indicators (home size, schools, and quality of life), suggesting the status quo is the same as the pre-dam level, if not worse. However, upstream-relocated participants show a positive perception of the indicators (2.44–3.18) driven by the availability of essential services (schools, health facilities, running water, electricity, marketplace, etc.) at the new settlement, which allows the settlers to flourish in their new social lives.

The same ranking analysis of the Merowe Dam's impact on living conditions in the resettlement areas classified by level of education and gender of participants showed insignificant differences. Once again, this demonstrated that the variation in Merowe Dam's social effect has touched communities' differently across location but within the same location they experience similar impact regardless of the level of education and gender analysis).

Spearman rank correlation analysis was conducted to test the correlation among social indicators related to the suitability of living in resettlement areas to determine if the perceptions of the communities on the effects are connected and reliable. Table 8 presented significant positive correlation amongst all three indicators, thus giving assurance on the reliability and robustness of communities' perception ranking across all districts.

**Table 8.** Correlation among indicators of "suitability of living" in resettlement areas.

| | Home Space and Design | Children Schooling | Quality of Life |
|---|---|---|---|
| Home Space and Design | 1.000 | | |
| Children Schooling | 0.558 | 1.000 | |
| | 0.000 | 0.000 | |
| Quality of life | 0.503 | 0.601 | 1.000 |
| | 0.000 | 0.000 | 0.000 |

Note: The second row shows *p*-values ($p < 0.001$). Source: Author's calculation.

The overall ranking of indicators in Table 8 shows that home size ranks highest and quality of life lowest. However, this ranking does not reduce the degree of satisfaction with resettlement. Local official interviewee states:

*"The social aspect before the dam was challenging. We used to live traditional lives in the villages, scattered across a wide area with no services and a community depending on the clan. There were 99 islands and no proper schools, no electricity or roads and the distance between villages were very far and we lacked transports. Our life at the new settlement is 100 times better in all aspects".*

Many local farmers respondents agreed with this position, with one suggesting:

*"The Merowe Dam provides many benefits to the country, especially to sons of the displaced. The benefits of education and health services and the future will be bright for them".*

Most upstream-relocated and upstream-resident participants agreed that the new settlement is better than the old villages, because of services and facilities for health, education, a marketplace, transport, etc. However, some expressed an emotional attachment to their old life while still having a positive perception of the new place to be suitable for living, as one local farmer stated:

*"The new place is suitable for living because it has services and health facilities. The new settlement is better because of health facilities and schools but the old one is better in term of living".*

The purpose of resettlement is to improve the living standards of dam-affected people beyond their pre-project levels. In this regard, the resettlement has supported the restoration of many aspects of the displaced lives, especially considering the agriculture scheme attached to each settlement, despite deficiencies in production. Figure 3 shows that there are adequate services in the settlement, which allowed the displaced to resettle with 60–98% of upstream-relocated communities believing that there is adequate electricity, health facilities, water, and schools. However, while 56% do not see an adequate urban center and 95% do not see adequate leisure centers, the water service is also not fully appreciated by the settlers.

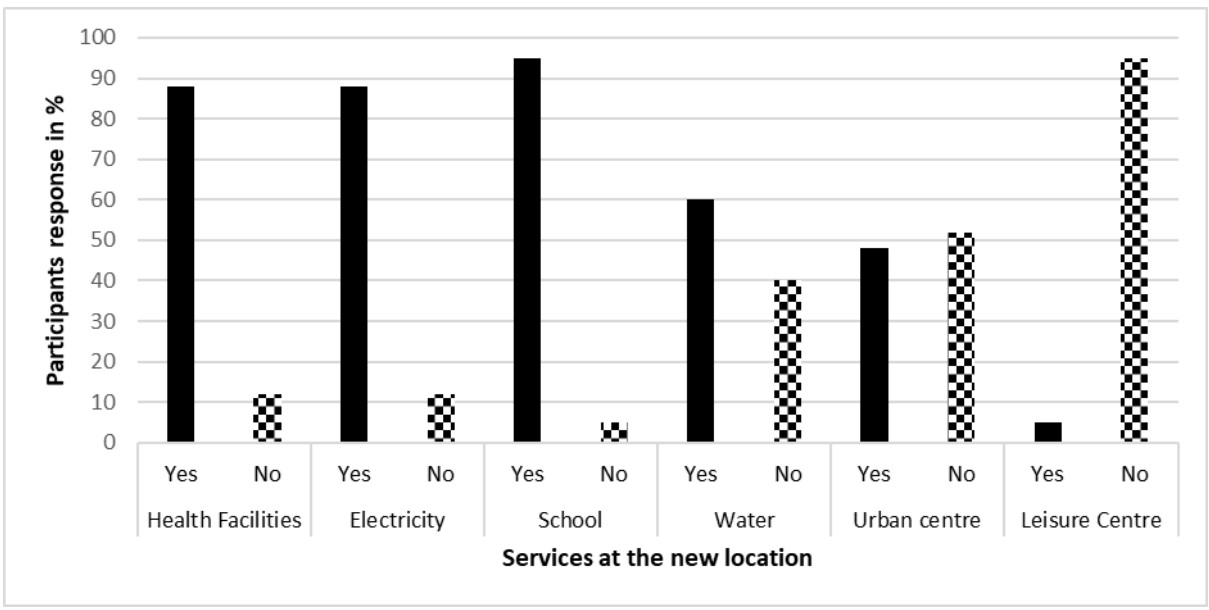

**Figure 3.** Adequate services at the new location: Source: Author.

This is due to many complaints about an irregular supply of irrigation water, and many settlers have been ill because of drinking from underground wells. A local official suggests:

*"The houses are generally very good compared to the old ones. They are suitable for us, but there is a drinking water problem. We used to drink from the Nile and now we drink from underground wells. We could not adapt to drinking underground water. However, now there are efforts to build a drinking water station at the Nile".*

Despite the high level of appreciation showed by respondents regarding houses and services at the resettlement sites, it is important to consider governmental bias in overstating the condition of services and houses. However, many observers, interviewees, and this research have observed huge improvements between the resettlements and pre-dam set-

tlements. Furthermore, many respondents, observers, and experts noted opportunities the dam brought for women, especially among upstream-relocated communities. Some participants suggest the resettlements have had a positive social influence on women, as many are freed from family farming duties because farms have been moved away from houses. However, this may have some negative consequences on farming economics and family income.

This allows women to engage in many social and other activities, such as development programs for women and increases in girls' school attendance because of accessibility: all schools are within walking distance. A local female interviewee elaborates on resettlements' influence on the female socio-economic position within the new social order at the resettlement:

> *"The relocation has positive and negative impacts for women, economically and socially. Socially, the resettlement has benefited women more in term of education and learning new skills like sewing, cooking, craft, etc. Also, it enables women to engage in public life and to be up to date with local, national and international affairs through the availability of media (e.g., TV). Furthermore, there are significant improvements in housework by using modern electronic equipment".*

Services are key factors in restoring displaced people's lives to pre-dam levels or better. Education has been a champion for many settlers. Furthermore, some local farmers and settlers perceive resettlement as a gateway to connect with the rest of Sudan and the world.

> *"Our settlement is close to main roads and provides our sons with a better environment for education and they excel in it. We are leading the region in the national results for primary school and high school exams. Our sons now use computers, the internet, and smartphones, allowing them to connect with the world. Now they are exposed to all kinds of sport, and we have football teams. I believe we will have a new generation who will be highly skilled in many fields. We have access to electricity, which allows us to have air conditioning, television, nice houses, mobile phones and everything else available to us"* (local female interviewee).

To understand whether settlers are satisfied and if the settlement has helped them to restore some of their former life, Table 9 shows a high degree of satisfaction for 77% of participants across two Merowe Dam settlements (Amri 2 and Amri 3). This is not only strong evidence of dam settlements' improvement but challenges the critics of dams who seek a global standardization of dam resettlement. Furthermore, to ensure that the displaced and relocated communities are helped to restore their lives, they must have access to resources. Most research focuses on displacement and resettlement schemes, such as infrastructure, compensation, housing, and livelihood support, ignoring the quality and quantity of natural resources and the time people spend accessing them, as well as their right to them. In the case of Merowe Dam, according to some interviewees, long-term livelihood support for resettled people was provided primarily through agriculture schemes, but also in fishing, jobs in manufacture, services, and access to national markets.

**Table 9.** Opinion of upstream-relocated residents about the new resettlement.

|  |  | Frequency | Percent |
|---|---|---|---|
|  | Dissatisfied | 23 | 23% |
| Valid | Satisfied | 77 | 77% |
|  | Total | 100 | 100% |

Source: Author questionnaire 2017.

## 7. Social Mobility, Infrastructure, and Services

Merowe Dam is similar to other dams in respect of some social impacts, but the main departure point is regarding social mobility. The results suggest that Merowe Dam and its supportive projects have increased social mobility through infrastructure development,

such as educational and health facilities, agriculture schemes, roads, bridges, and airports, all of which have a positive socio-economic influence by easing movement of goods and people across Merowe region and the country. This infrastructure led to increased tourism in the region because many historical sites are in the area, most importantly Jabal Al-Barkal, providing further socio-economic and economic development with increases in social mobility, income, and jobs in the region.

Before the Merowe Dam, the area was hostile and only habitable on the fertile strip of land adjacent to the Nile. This led many northerners to migrate to central Sudan seeking a better life. Sudanese communities are interconnected and have rural and urban extensions reflected at the community level due to migration, particularly from northern Sudan communities (Haberlah 2012). Post-Merowe Dam social lives in the region have greatly expanded and become more diverse, which is driven by a huge influx of migrants because of socio-economic development activities, especially services. To reduce the social tension, enhance socio-economic development, and improve the resettlement communities, the Sudanese government has also initiated numerous accompanying economic and social service projects alongside Merowe Dam. For example, a local farmer explained:

> *"As a Northerner, migration is embedded in our blood. Seeking better life opportunities, especially when it comes to education. We do everything to educate our children. Before the dam, many children did not attend school, especially girls, due to the distances between villages and schools. Yet, Merowe Dam and its accompanying projects have given us the means to settle into the region with good services, especially education. Now we have very good schools and six colleges and universities, for example, Merowe University of Technology Abdulatif Alhamad. The availability of education at all levels across the region, including the new settlement, reduced the cost of education. Also, it allows families to stay in the region and get the best education without moving to Khartoum or sending their sons away".*

The population of downstream and resettled communities has increased because of migration from across Sudan. This finding points to social improvements in the region through immigration, which led to diversification of the social fabric in the region. Likewise, easy access to the region has contributed significantly to the region. Before the initiation of these projects, it took four days to reach Merowe from Khartoum, but now, with luxury high-speed buses operating in the region, it takes just 3–4 h. With respect to social mobility and socio-economic changes in the region, local committee members added:

> *"A project that does not increase the population density in the region will not succeed, even if it has all the services available. The Merowe Dam attracted people from all regions of Sudan: workers, employees, investors, and traders. This is the most important element driving stability, reducing migration from the region, and providing better living standards for the displaced and locals".*

Certainly, there is some improvement in the urban center of Sudan in term of services. The Merowe Dam has contributed immensely to that, especially in providing electricity. However, the scope of this research is regional, particularly the rural communities of the Merowe region, in comparison to other rural areas in northern, eastern and Blue Nile states of Sudan (except Darfur and Nubba Mountain because of conflict). The Merowe Dam and its accompanying projects have made a huge difference in the Merowe region. The difference is noticeable compared to other regions with respect to services, access to electricity and overall socio-economic development.

## 8. Conclusions and Policy Implications

The study explores the lesser challenged and most untouched matter associated with the distribution of mega-dams' social effects on the hosting societies. This paper examined the perception and understanding of hosting communities about these effects since Merowe Dam's main aims are to improve communities' social and economic conditions in the region. Therefore, the success of achieving these goals, to a large extent, depends on the provision

of suitable resettlement and mitigation of displacement impact by restoring communities' lives by providing means of stable livelihood and better services (electricity, schools, health facilities, and market place) for both displaced and local communities. The findings suggest that resident's (resettled and displaced alike) are clearly aware of the progressive effects of Merowe Dam schemes have over the region's social mobility. Nevertheless, the degree of this awareness is still limited and surrounded by the intangible effects, which are related directly to their feeling about loss, nostalgia to their history, culture and grievances, and their sources of revenue which have a direct relation to how different social indicators were ranked by the communities.

Communities at upstream-relocated and upstream-resident ranked low for indicators, such as, 'archaeological and historical sites', 'feeling of belonging to the place', and 'self-esteem', even though these indicators were identified by communities as the main affected areas. Additional examination of applicable indicators related to adequate services at the new location such as electricity, schools, health facilities, access to drinking water, and urban center, confirmed and reinforced the perceptions of residents regarding Merowe Dam's positive social effects. The analysis of residents' perception is not usually considered as hardly equally significant as cost–benefit assessments. In this paper, community perception was considered as complementary to the conventional cost–benefit analysis and therefore signifies a valuable tool for evaluating social impacts of mega-dams.

Nevertheless, the perceived strength of the residents drops abruptly on a few social-indicators with sentimental values, which appear to harm communities in the region. For example, the perceived ill-treatment by the government was the key contributor in promoting lower ranking in some social indicators. Thus, solving the dispute over the relocation package with Manasir communities and the sentiment of ill-treatment among affected communities through an effective and reliable emotional and livelihood system of support and monitoring at the local and regional level for the displaced and resettled communities is needed. The residents are not only mindful of the significant role of services provided through Merowe Dam's complementary projects and infrastructure that support social progress in the region but also for the whole population as the region of Merowe was largely isolated before the dam.

The social influence of Merowe Dam in rural Merowe is embedded in frequently perceived global concerns over the negative impacts of dams on host communities (Ersumer 1999). However, this paper has explained the extent to which Merowe Dam has improved or damaged social aspects of the local communities. It is clear from the results that social impact is embedded within the economic input of Merowe Dam in the region, which is interlinked with the overall socioeconomic changes that occurred in the region. This research indicates that, without links to advanced infrastructure and services, accomplishment of social improvements may not have been possible in rural Merowe communities. The study clearly revealed that communities viewed Merowe Dam as a powerful new opportunity to overcome the disadvantages and stagnation in social progress caused by isolation of the region, which has implications in understanding the relationship between development and remote location on everyday life of communities residing in such regions. This research clearly revealed that Merowe Dam's positive social influence outweighs negative ones in the Merowe region.

The following policy implications can be drawn from the results of this study. Long-term monitoring of social impacts of resettlement and improvements needs to be established. This could influence the planning and implementation of resettlement programs effectively and mitigate negative social impacts. As Scudder highlighted, with resettlements in developing economies, dams failed due to a lack of long-term support and treated as a one-off activity instead of a continuous process. Long-term monitoring of resettlements would be a significant policy shift in supporting displaced people, to restore their social lives, socio-economic status, and well-being. In addition, there is a need to strike a balance between enhancing social improvement and livelihood of local communities

and agribusiness through efficient utilization of water, electricity, and other infrastructural facilities provided by the Merowe Dam.

## 9. Limitations and Future Research

Women's contribution in this study was limited, due to cultural and traditional restrictions where women's involvement in public and political affairs is limited. The Sudanese society, mainly dominated by males in public affairs and access to housewives and/or female members of the family, has to be through male members—which was adopted in this study. Therefore, having an independently drawn viewpoint from the female members only without the presence of any male members could have provided enhanced understanding of the outcome of the research. Therefore, further in-depth socioeconomic research on the impact of Merowe Dam on women and their contribution to household economy is recommended. This may provide sufficient details on how the dam changed the socioeconomic condition of women at the household and community level.

Furthermore, the present research was conducted under an authoritarian regime where freedom of speech is limited. Therefore, participants were not able to express their opinion openly. The researcher had to follow the guidelines provided by the authorities, which implied limited freedom to conduct in-depth interviews. Hence, an in-depth political ecology study may provide more details on the socioeconomic, cultural, environmental, and political influences of the Merwe Dam at the community level. The present study did not focus on the political influence of the Merowe Dam in details. The expectation is that future regimes in Sudan may provide a better and open environment for conducting such research with no limits on freedom of speech.

**Supplementary Materials:** The Supplementary Materials are available online at https://www.mdpi.com/article/10.3390/economies9040140/s1.

**Author Contributions:** Conceptualization, A.-N.A. and S.R.; methodology; A.-N.A.; software, S.R.; validation and formal analysis, A.-N.A. and S.R.; investigation, A.-N.A.; resources, A.-N.A.; data curation, A.-N.A.; writing—original draft preparation, A.-N.A.; writing—review and editing, S.R. and A.-N.A.; supervision, S.R.; project administration, A.-N.A.; funding acquisition, A.-N.A. All authors have read and agreed to the published version of the manuscript.

**Funding:** This research received no external funding it was self-funded research by the main researcher Al-Noor Abdullah.

**Institutional Review Board Statement:** The study was conducted according to the guidelines of the declaration of Science and Engineering Research Ethics Committee School Representatives of University of Plymouth, Research Ethics Committe and date of approval 6 February 2017.

**Informed Consent Statement:** Informed consent was obtained from all participants involved in the study.

**Acknowledgments:** I would like to express my gratitude to my primary supervisors, Sanzidur Rahman and Geoff Wilson, who guided me throughout this project. I would also like to thank my friends, family, and the participants who supported me and offered deep insight into the study.

**Conflicts of Interest:** The authors declare no conflict of interest.

## Appendix A
### Households Questionnaire Survey
### Local Participants Downstream and Upstream

Assessing mega-dam economic contribution and social, political, and environmental issues in development context.
### Introduction

The purpose of this questionnaire is to solicit your opinion and perception about large dams and their role in the development of a nation, particularly Sudan. This will be looked at within the context of the Merowe Dam project. This interview into large dams is part

of research towards a Doctor of Philosophy Degree at the University of Plymouth UK. I therefore thank you for sparing some of your time to discuss this topic with me.

**Basic background information**

Participant ID: ——————————————————————————————

**Q1-** Name of Village / Community / Settlements: ——————————————————

| Q2- Age of participant | | |
|---|---|---|
| **Q3-** Gender | Male | |
| | Female | |
| **Q4-** Level of education | | |
| **Q5-** Number of household members (including children) | | Male |
| | | Female |

| Q6- What type of farming do you do? (please order in terms of importance) | Date palm trees | |
|---|---|---|
| | Vegetables | |
| | Arable crops | |
| | Livestock | |
| | Others (If others, please specify) ———————— | |

| Q7- What proportion of your total income is derived from these sources (please provide percentage values) | Farming | % |
|---|---|---|
| | Fishing | % |
| | Trading/ Selling | % |
| | Others (If others, please specify) ———————— | % |

**Merowe Dam**

**Q8-** How do you feel about the presence of the dam? ————————————————
——————————————————————————————————————————
——————————————————————————————————————————
——————————————————————————————————

**Q9-** What do you think are the main purposes of Merowe Dam? ——————————————
——————————————————————————————————————————
——————————————————————————————————————————
————————————————————————————————————————-

| Rate the following indicators (1–5) based on observed influence to you since the construction of Merowe Dam (1 is Strongly disagree, 2 Disagree 3 Neutral, 4 Agree and 5 Strongly agree) | Strongly disagree | Disagree | Neutral | Agree | Strongly agree |
|---|---|---|---|---|---|
| **Q10- Economic** | | | | | |
| a.　Irrigation of agriculture | 1 | 2 | 3 | 4 | 5 |
| b.　Industrial development in the region | 1 | 2 | 3 | 4 | 5 |
| c.　Employment generation | 1 | 2 | 3 | 4 | 5 |
| d.　Infrastructure (e.g., bridges, roads, hospital, schools) | 1 | 2 | 3 | 4 | 5 |
| e.　Access to electricity | 1 | 2 | 3 | 4 | 5 |
| f.　Communication network (e.g., telephone, internet) | 1 | 2 | 3 | 4 | 5 |
| g.　Transportation across the region (e.g., travel) | 1 | 2 | 3 | 4 | 5 |
| h.　Sources of income (e.g., market and working places) | 1 | 2 | 3 | 4 | 5 |
| i.　Cost of electricity | 1 | 2 | 3 | 4 | 5 |
| j.　Price of assets (e.g., business, land, stocks) | 1 | 2 | 3 | 4 | 5 |

| **Q11- Economic (farming)** | | | | | | |
|---|---|---|---|---|---|---|
| a. | Types of farm production | 1 | 2 | 3 | 4 | 5 |
| b. | Volume of farm production | 1 | 2 | 3 | 4 | 5 |
| c. | Grazing area for animals | 1 | 2 | 3 | 4 | 5 |
| d. | Land size | 1 | 2 | 3 | 4 | 5 |
| e. | Quality of the soil | 1 | 2 | 3 | 4 | 5 |
| f. | Cost of irrigation | 1 | 2 | 3 | 4 | 5 |
| g. | Price of farm products (e. g. date, vegetables) | 1 | 2 | 3 | 4 | 5 |
| **Q12- Social** | | | | | | |
| a. | Home space and structure | 1 | 2 | 3 | 4 | 5 |
| b. | Family ties and social relations | 1 | 2 | 3 | 4 | 5 |
| c. | Community coherence | 1 | 2 | 3 | 4 | 5 |
| d. | Social status | 1 | 2 | 3 | 4 | 5 |
| e. | Farm land as measure of wealth and pride (e.g., date trees) | 1 | 2 | 3 | 4 | 5 |
| f. | Archaeological and historical sites | 1 | 2 | 3 | 4 | 5 |
| g. | Feeling of belonging to the place | 1 | 2 | 3 | 4 | 5 |
| h. | Mental health | 1 | 2 | 3 | 4 | 5 |
| i. | Personal Health | 1 | 2 | 3 | 4 | 5 |
| j. | Self-esteem | 1 | 2 | 3 | 4 | 5 |
| k. | Children schooling | 1 | 2 | 3 | 4 | 5 |
| l. | Quality of life (work, schools water, electricity, etc.) | 1 | 2 | 3 | 4 | 5 |
| **Q13- Environment** | | | | | | |
| a. | Spread of Waterborne diseases (e.g., bilharzias) | 1 | 2 | 3 | 4 | 5 |
| b. | Habitats communities along the river bank | 1 | 2 | 3 | 4 | 5 |
| c. | Wildlife, | 1 | 2 | 3 | 4 | 5 |
| d. | Forests | 1 | 2 | 3 | 4 | 5 |
| e. | Water Logging | 1 | 2 | 3 | 4 | 5 |
| f. | Salinisation | 1 | 2 | 3 | 4 | 5 |
| g. | Erosion (wearing of soil, dirt, rock by water force) | 1 | 2 | 3 | 4 | 5 |
| h. | Sedimentation | 1 | 2 | 3 | 4 | 5 |
| i. | Water supply | 1 | 2 | 3 | 4 | 5 |
| j. | Water flow | 1 | 2 | 3 | 4 | 5 |
| k. | Distance to the river bank | 1 | 2 | 3 | 4 | 5 |
| l. | Flooding | 1 | 2 | 3 | 4 | 5 |

**Q14-** Please can you rank up to three the most important benefits of Merowe Dam to you? ————————————————————————————————————
————————————————————————————————————————————————
——————————————————————————————————————

**Q15-** Please can you rank up to three the most important three benefits of Merowe Dam to your community? —————————————————————————————————
————————————————————————————————————————————————
———————————————————————————————

**Q16-** What contributions do you think Merowe Dam has made to the development of the region since its construction? ———————————————————————————
————————————————————————————————————————————————
————————————————————————————————————————————————
———————————————————————————————————————————

**Q17-** Have economic matters, identified above in question 10 and 11 have been addressed? ————————————————————————————————————————
————————————————————————————————————————————————
————————————————————————————————————————————————
————————————————————————————————————————————————
————-

**Q18-** Do you think the perceived social issues identified in question 12 above have been addressed. ——————————————————————————————————————
————————————————————————————————————————————————

_________________________________________________________________

_________________________________________________________________

**Q19-**Has environmental issues identified in question 13 above been addressed? —————

_________________________________________________________________

_________________________________________________________________

_________________________________________________________________

_________________________________________________________________-

**Downstream**

**Q20-** Since building of the dam has your land area changed in size.

| **Q21-** How much land do you have (feddan)———— | | **(a)** Irrigated | Rain-fed | **(b)** What was your land size before the dam? | **(c)** How much extra land do you irrigate after building Merowe Dam? | |
|---|---|---|---|---|---|---|
| | | | | | Before | After |
| Own | | | | | | |
| Rent | | | | | | |

      Yes ☐         No ☐

**Q22-**How the change in water regime affect you farm production? ————————

_________________________________________________________________

_________________________________________________________________

________________________________________________

**Q23-**Is there an increased in the cost of irrigation because of the dam?————————

_________________________________________________________________

_________________________

**Q24-**Has the change in land size affected your income?

      Yes ☐         No ☐

**Q25-**To what extent if yes has, the dam (elaborate in percentages) affected the level of your income?————————————————————————————

_________________________________________________________________

_________________________-

**Upper-Stream**

**Q26-**Please explain why you are still living here by the reservoir?————————

_________________________________________________________________

________________________________-

**Q27-**Have you received compensation for properties lost (e.g., properties, houses, land, agriculture and agricultural products (palm and fruits and livestock))? —————

_________________________________________________________________

_________________________________________________________________

_________________________________________________________

**Q28-**How much did you receive? ————————————————————

________________

**Q29-**Were you satisfied with the compensation you received? ————————

_________________________________________________________________

_________________________________________________________________

________________________________________

**Q30-**Were you been given any other form of support in addition to financial (e.g., counselling, mental therapy)?

      Yes ☐         No ☐

**Q31-**Can you specify the nature of the support you received? ————————

_________________________________________________________________

_________________________________________________________________

_______________________________________________________-

**Q32-** Which organisation(s) or institution(s) where involved in supporting you? —————
______________________________________________________________________
______________________________________________________________________
_________________________________________________

**Q33-** In your opinion was there a lack of support? ——————————————
______________________________________________________________________
___________________________________________-

**Q34-** If yes could please explain why there was a lack of support? ————————
______________________________________________________________________
______________________________________________________________________
__________________________________________________

**Q35-** To what extent are you satisfied with new settlement compared to your old place? ——
______________________________________________________________________
______________________________________________________________________
______________________________________________________________________
________________________________________________

**Q36-** Does the new location have adequate services such as (select all relevant option)?
　　Health facilities　□　　　　Electricity　□　　　Schools　□
　　Leisure centre　□　　　Water　□
　　Urban Centre (market and work place)　□

**Q37-** What is your assessment of the resettlement location as compared to your old place in terms of facilities (e.g., health, schools water, electricity)? ——————————
______________________________________________________________________
______________________________________________________________________
_____________________________________________________-

**Q38-** To what degree has the distance changed between the new farm and your home compared to how it was before the dam, (please explain in kilometres)? ——————————
______________________________________________________________________
______________________________________________________________________
____________________________________________

**Q39-** How has the change in distance affected your life? ———————————————
______________________________________________________________________
______________________________________________________________________
________________________________________________-

**Q40-** Has building the dam caused any conflict of land right in the old settlement alongside claims in the new settlement?
　　Yes　□　　　　　No　□

**Q41-** If yes, how serious is the conflict? ————————————————————
______________________________________________________________________
______________________________________________________________________
________________________________________________-

**Q42-** In your opinion has building the dam caused any social conflict amongst communities (please explain)? ——————————————
______________________________________________________________________
______________________________________________________________________
________________________________________________-

**Q43-** How about your relation with friends and relatives at old community? ——————
______________________________________________________________________
______________________________________________________________________
______________________________________________________________________
______________________________________________________________________

**Q44-** Do you know any household in your village/community have left the area completely because of the dam? ——————————————————

_______________________________________________________________

_________________________________________________

**Q45-** Have you or a member of your family been ill because of waterborne disease caused by building of the dam?

Yes ☐ No ☐

| **Q46-** If yes, what kind of common diseases do you or your household suffer from | | | **(a)** the frequencies | **(b)** What do you think are the main reasons? |
|---|---|---|---|---|
| | Before 2004 | After 2004 | | |
| Malaria | | | | |
| Cholera | | | | |
| Bilharzias | | | | |
| Typhoid | | | | |
| Chest problems | | | | |
| Other | | | | |

**Q47-** In your opinion have you received a fair share of the economic benefits of the dam (please explain)?—————————————————————————————

_______________________________________________________________

_________________________________________________

**Q48-** How about your community have they received a fair share of the economic benefits of the dam (please explain)?—————————————————————————

_______________________________________________________________

_______________________________________________________________

______________________________________________________-

**Q49-** Was there a period of consultation with you before the construction of Merowe Dam?

Yes ☐ No ☐

**Q50-** If your answer is yes, what was the nature of the consultation? ———————————

_______________________________________________________________

_______________________________________________________________

___________________________________________

**Q51-** What do you think about the duration of consultation? ————————————

_______________________________________________________________

_______________________________________________________________

_____________________________________________-

**Q52-** Was the consultation process (if any) open (please explain)? ————————————

_______________________________________________________________

_______________________________________________________________

_______________________________________________________

**Q53-** Was the consultation process (if any) involve everyone in your community (please explain)? ————————————————————————————

_______________________________________________________________

_______________________________________________________________

________________________________________________________-

**Q54-** In your opinion was Merowe Dam project a result of consensus of local/regional stakeholders? ————————————————————————————

_______________________________________________________________

_______________________________________________________________

________________________________________________________-

**Q55-** What do you think of the on-going opposition from many of (Hamadab, Amri and Manasir communities) against Merowe Dam? ————————————————

_______________________________________________________________

_______________________________________________________________

_______________________________________________________________

_______________

**Q56-** Who do you think are the key decision makers in Merowe Dam development? ———
————————————————————————————————————————
————————————————————————————————————————
——————————————————————————————————————

**Q57-** What role do you think political influences play in the decision to build Merowe
    Dam? ———————————————————————————————————
————————————————————————————————————————
————————————————————————————————————————
————————————————————————————————————————
——————————————-

**Q58-** Do you think there could have been alternatives to building Merowe Dam?
    Yes  ☐            No  ☐
**Q59-** If yes, can you give examples? ——————————————————————
————————————————————————————————————————
————————————————————————————————————————
——————————————————————————————————-

**Q60-** Finally, what is your opinion on the overall building of Merowe Dam? ——————
————————————————————————————————————————
————————————————————————————————————————
————————————————————————————————————————
—————————————————————————————————————

Thank you for taking some of your time to answer the questions. Please feel free to add any comments that you may have.

**Appendix B**

**Semi-structured interview institutional actors**

Assessing mega-dam economic, political contribution, and social and environmental problems in development context.

| Main Questions | Additional Steering Questions | Clarification Questions |
| --- | --- | --- |

**Introduction**

Explain who I am, my department, and my research.
Explain why I am doing the research—take no sides, merely investigating.
What I hope to achieve, how and why.
Give copy of information for participants' sheet.

**Ethics**

Explain why recording and making notes, request permission to record, explain option to stop recording/say no, etc., at any time. Reassure re privacy and confidentiality. Request signature on consent form. Leave copy of info for interviewees sheet with participant. Sheet includes contact details should they wish to contact me.

**Section 1 Basic information**

1.  Name of Interviewee: ———————————
2.  Title: ———————————-
3.  Gender: ———————————-
4.  Position: ——————————————————————.
5.  Name of Organisation ————————————————

**Section 2 Development**

1-What comes to your mind when the word development is mentioned?
2-What is your view on mega-project as mean of achieving development of a country?
3-What do you make of dams' role as development strategy tool?
4-How important is the Merowe Dam to the development of Sudan?
5-Do you think the construction of Merowe Dam was necessary?
6-In your opinion what is the purpose of build Merowe Dam?

**Section 3 Economic**

**1-In your opinion did Merowe Dam make a significant economic contribution locally and nationally?**

- In your opinion what are the main economic contribution of the dam for the communities?
- How do you think the communities are benefiting from it?
- In your opinion what change has the dam brought to socio-economic status of communities?
- Do you see any unforeseen economic cost for the dam?

**2-How do you view its impact on irrigation agriculture in supporting economic growth?**

- In your opinion, does building of Merowe Dam increase food production in the region?
- Can you explain irrigation scheme set by the authority?
- Is it working for the benefit of the communities?
- Is there any change in products prices?
- How about the income of the communities?

**3-Does the building of the dam help in any way in creating employment opportunities in the region?**

- Can you explain in numbers and sectors?
- What type of jobs it brings to the communities?
- Are these jobs relating to the dam or other sectors?

**4-From my understanding of industrial development is one of the key objectives of Merowe Dam. In your opinion has this objective been achieved.**

- What type of industry was developed?
- Where these industries built?
- Has the dam helped communities to have access to jobs within these industries?

**5-Do you think Merowe dam has provided long-term energy security?**

- How do you see communities accessing electricity?
- Is the electricity affordable to people in the region?
- Do you think local communities have benefited from electricity generated by the dam?
- What impact electricity has on the people life?
- Are there any changes in the price of electricity?

**Section 4 Social**

**1-How do you think displacement and resettlement affected the communities?**

- What are the most important social aspect that affected by the displacement?
- To what extent do you think displaced people have been supported by the government?
- Was there any conflict between communities as result of displacement and settlement?
- What is your view on the new settlements?
- Was it adequate in term of facilities and services?
- In comparison to the old settlement and new settlement, which one is better?
- Could displacement have been avoided?
- Do you see any social benefit of the dam for the communities?

**2-Does Merowe Dam cause restriction to resource access or any land disputes?**

- What about the land right in new and old settlement?
- Was there any plan for people to access their old land?
- Are there any tribal disputes on land or territories?

**3-Has there been at any time health concern during the dam life cycle?**

- In your opinion what are the main area of health concerns?
- Is here any illness detected?

- What is the cause of the illness?

**4-Do you think the dam has impact on the rural community fabric?**

- In which way do you think it has impacted communities the most?
- Has the dam divided families because of displacement and relocation?
- How this impact communities' relation?

**5-In your opinion has Merowe Dam generated any kind of social conflict amongst communities, societies, NGOs, and the authorities?**

- What are the reasons behind the conflict?
- How these conflicts affected the relationship between these groups?
- Do you think the issues caused conflicts to be addressed?
- In which way have they been addressed?
- What is your opinion on the demonstration against the dam by many communities?

**Section 5 Political**
**1-In your opinion was building Merowe dam is a result of state political influence?**

- Do you think there is a link between mega-projects and politics?
- Who do you think have high stake in decision making in building the dam?
- Do you think the selection of Merowe area is right?
- Was there any bias in selecting Merowe area?

**2-What is your though about funding of Merowe Dam?**

- What is the source the fund?
- How this impact on the control over the dam?
- How do you think the financial sanction against the country impacted the delivery of the project?
- Was there any concession from the government to secure the money for project?

**3-For the project to go ahead do you think the government has changed its development politics to suite building dam?**

- For example, is there any change in regional development scheme in the country?
- Was there any movement toward relaxing environmental policies because of the dam?

**4-Was the process of compensation for displaced people fair?**

- How was the process?
- Based on what method the loss properties compensated?
- Do you think displaced people were satisfied with compensation?
- What you think should be done differently?
- Do you think the ongoing opposition to the dam means dissatisfaction?

**5-Was there a consultation process in building Merowe Dam?**

- When the consultation started?
- How long the duration of the consultation?
- In your opinion did the consultation include all affected stakeholders.
- To what extent you think the outcome of the consultation made a different in decision-making.
- Was the consultation open and transparent?

**Section 6: Environmental**
**1-Dams have been controversial when it comes to environment. What is your opinion?**

- Do you think these dams are only damaging environment?
- How can this assumption change?
- Do you think there is a deliberate intention to demonised dam?
- In the age of growing energy demand do you see the dam as solution?
- Can water management project (dams) play a role in combating climate change?

### 2-How Merowe Dam impacts ecosystem both upstream and downstream?

- What impact the dam has on fisheries
- In your opinion, which side of the dam affected the most?
- Is there difference in type and degree of impacts?
- Is any ecological benefit can being drive from building the dam?

### 3-To what extent in your opinion sedimentation issue has become a problem?

- How do you see the sediment concentration in the reservoirs?
- What impact it may have on irrigation canals?
- How this issue can be resolve?

### 4-What are the effects of Merowe Dam on the river system and water supply?

- Has the dam affected water supply?
- Is there any change in water flow?
- Has the change affected irrigation system downstream?
- Do you see any benefit from the new water regime?
- How the change effect ecosystem downstream?

### 5-Dams are associated with salinisation and waterlogging, what is your view on this perspective?

- As observer in this field, how you see dam in relation to these issues?
- How salinisation and waterlogging impact the quality of soil?
- Does this problem impact the level of productivity of the land?
- Is there any way of reducing the impact of salinisation and waterlogging?

### Finally, what do you think of the dam overall?

Thank you very much for your time and contribution to this study.

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
