# Peer review of "Social Impacts of a Mega-Dam Project as Perceived by Local, Resettled and Displaced Communities: A Case Study of Merowe Dam, Sudan"

_economies, doi:10.3390/economies9040140_

Round 1

Reviewer 1 Report

Dear Authors,

You have completed a detailed study of the social impacts of one mega dam in Sudan.

It is a very rich snapshot of a society in transition and it has been expertly documented, analysed and discussed. I particularly admire the way that both positive and negative impacts of the dam project were questioned, collated, evaluated, incorporated into the analysis and discussed.

On page 4 you mention that Jalon (1994) noted the ecosystem effects of dams and consequently you mention that not only social and economic impacts are to be considered but that environmental impacts are also of importance in these types of analyses (Power, et, al. 1996). Even though your survey questions were designed to collect data on the environmental impacts, no findings with respect to this were found. Even if the impact of environmental factors on the social and economic aspects was entirely neutral, it would be a significant finding. Could the authors please indicate some level of findings with respect to changes to the environment and the impact on the communities surveyed?

I note that the numbers of women in the survey were much less than men and that a cultural explanation of the reasons for that was given. A number of claims were made throughout the paper with respect to the answers being the same, regardless of gender. Would the authors please reflect on the conditions under which females were interviewed and whether there may have been a dominant male present during the interview? Could this have influenced the answers given and is this a possible limitation of the study?

Could the authors please reflect upon and document the limitations of this study more generally and how it could have been improved if the opportunity were presented again?

A minor issue with regards to figure 2.2. Generally, an equation does not require its own figure number and caption. It only needs an equation number and reference to that equation number in the accompanying text.

It is significant that the authors have noted the longer-term intangible effects on the community and the need for community support over the longer term. Well done.

Kind regards.

Author Response

Dear Authors,

You have completed a detailed study of the social impacts of one mega dam in Sudan.

It is a very rich snapshot of a society in transition and it has been expertly documented, analysed and discussed. I particularly admire the way that both positive and negative impacts of the dam project were questioned, collated, evaluated, incorporated into the analysis and discussed.

RESPONSE: Thank you very much for your complementary comments.

On page 4 you mention that Jalon (1994) noted the ecosystem effects of dams and consequently you mention that not only social and economic impacts are to be considered but that environmental impacts are also of importance in these types of analyses (Power, et, al. 1996). Even though your survey questions were designed to collect data on the environmental impacts, no findings with respect to this were found. Even if the impact of environmental factors on the social and economic aspects was entirely neutral, it would be a significant finding. Could the authors please indicate some level of findings with respect to changes to the environment and the impact on the communities surveyed?

RESPONSE: In response to your comments in relation to the environment. I have addressed it by adding some of the findings on the text from lines 353 – 363. The article is solely focused on the socioeconomic impact however the main study has a chapter on political and environmental impact.

I note that the numbers of women in the survey were much less than men and that a cultural explanation of the reasons for that was given. A number of claims were made throughout the paper with respect to the answers being the same, regardless of gender. Would the authors please reflect on the conditions under which females were interviewed and whether there may have been a dominant male present during the interview? Could this have influenced the answers given and is this a possible limitation of the study?

RESPONSE: The point has been elaborated and explained in lines 302 – 306. I have addressed this point from a social structure and cultural viewpoint as a Sudanese myself am aware of the context of the country and the research.

Could the authors please reflect upon and document the limitations of this study more generally and how it could have been improved if the opportunity were presented again?

RESPONSE: A limitation section is added at the end of the manuscript which focused on the limited participation of women (Section 8).

A minor issue with regards to figure 2.2. Generally, an equation does not require its own figure number and caption. It only needs an equation number and reference to that equation number in the accompanying text.

RESPONSE: I have changed the equation, deleted the caption and kept it as equation 1.

It is significant that the authors have noted the longer-term intangible effects on the community and the need for community support over the longer term. Well done.

RESPONSE: Thank you very much for your complementary comments. We believe that the revisions made to accommodate your suggestions is satisfactory and the present revised version is suitable for publication in the journal.

Kind regards.

Reviewer 2 Report

Thank you for the opportunity to read the paper “Social impacts of a mega-dam project as perceived by local, resettled and displaced communities: A case study of Merowe Dam, Sudan”.

Please see below my comments on this paper.

Introduction

„Many areas of mega-dam impacts were investigated including social impacts, environmental aversion, political corruption, governmental connections and philosophy of undertaking projects instigating nationalism.” – please provide some references.

I suggest the author(s) to restructure the theoretical background by dividing it into two chapters, Introduction and Literature review.  

I suggest the authors to remove the demographic data from the chapter Results and include it into a new subchapter .

Methodology and Results

The authors should present in more detail the structure of the questionnaire.

„The study used qualitative and quantitative methods (...) Moreover, in-depth interviews were conducted (....). Furthermore, two focus group discussions (....) Nevertheless, both interview techniques: in-depth and semi-structured interviews have disadvantages...  - Did the authors use focus group discussions or  semi-structured interviews?

Second, the results were not clearly presented. The authors combined quantitative and qualitative research methods, both among locals and among local decision makers. They also used two qualitative research methods (i.e. in-depth and semi-structured interviews). However, the results of the research include, combined, all these methods. I suggest the authors to restructure the Results chapter, presenting the results obtained with each method (i.e. the results of qualitative research - in-depth and semi-structured interviews - and the results of quantitative research). I also suggest that authors separate the results of quantitative research that include answers to open-ended questions from those that include answers to closed-ended questions (with answer options).

I suggest the authors to add a final chapter, Limitations and Future research.

I wish the authors all the best!

Author Response

Thank you for the opportunity to read the paper “Social impacts of a mega-dam project as perceived by local, resettled and displaced communities: A case study of Merowe Dam, Sudan”.

Please see below my comments on this paper.

RESPONSE: Thank you very much for your insightful comments. We undertook a major revision in response to most of your comments and also explained when it is not appropriate to change drastically. Details are provided below.

Introduction

„Many areas of mega-dam impacts were investigated including social impacts, environmental aversion, political corruption, governmental connections and philosophy of undertaking projects instigating nationalism.” – please provide some references.

RESPONSE: References have been added into the section you have pointed out.

I suggest the author(s) to restructure the theoretical background by dividing it into two chapters, Introduction and Literature review.  

RESPONSE: The introduction has been divided into two chapter as suggested. Furthermore, the introduction has been improved as you requested

I suggest the authors to remove the demographic data from the chapter Results and include it into a new subchapter.

RESPONSE: The demographic data have been removed and put as a subsection 4.1 the demographic characteristics of the study areas.

Methodology and Results

The authors should present in more detail the structure of the questionnaire.

„The study used qualitative and quantitative methods (...) Moreover, in-depth interviews were conducted (....). Furthermore, two focus group discussions (....) Nevertheless, both interview techniques: in-depth and semi-structured interviews have disadvantages...  - Did the authors use focus group discussions or  semi-structured interviews?

RESPONSE: The study used both semi-structured interviews and two focus groups for the overall PhD thesis. However, the information presented in this paper did not necessarily come from the outcome of the focus group discussions. Therefore, to avoid any confusion, the element of the focus group discussion as a research tool has been removed from the data collection methods section in this paper. This is because the outcome of the focus group discussions were not included anywhere in the paper. The main thesis however contains information from focus groups. Also, regarding the issues of the in-depth and semi-structured interview, the issue has been clarified by amending the line. Actually, in-depth interviews were conducted using a semi-structured interview technique or questionnaire. We have added the Semi-structured interview outline (Please see Appendix B) and also a transcript of one completed interview is included as supplementary material.

Second, the results were not clearly presented. The authors combined quantitative and qualitative research methods, both among locals and among local decision makers. They also used two qualitative research methods (i.e. in-depth and semi-structured interviews). However, the results of the research include, combined, all these methods. I suggest the authors to restructure the Results chapter, presenting the results obtained with each method (i.e. the results of qualitative research - in-depth and semi-structured interviews - and the results of quantitative research). I also suggest that authors separate the results of quantitative research that include answers to open-ended questions from those that include answers to closed-ended questions (with answer options).

RESPONSE: The total study is huge with many information which are interesting as this is a PhD project of the first author. The paper is organised under themes. And an integrated approach to present all materials collected through different methods relevant to prove the conclusions drawn on the theme was adopted in this paper as well as for the main PhD thesis. If results from each method of data gathering is reported separately, then those will become standalone findings but the integration aspect will be lost which is essential in order to draw concrete conclusion on the theme, e.g., displacement impact theme. Results on this theme is presented in an integrated form using response from the 300 survey respondents and selective interview responses which covered details on this theme. The results are also discussed within the context of existing literature on the theme. In fact reporting data from in-depth interviews was very selective to keep the article within an optimum length. However, in order to satisfy your suggestion, we have provided the generic “Semi-structured Interview questions” in the Appendix B. We have also added one complete transcript of in-depth interview as supplementary material. We believe that readers who are interested to know what line of questioning was followed in the interviews and the depth of answers received can learn this by browsing the appendix section and the supplementary material file. The main survey questionnaire (Appendix A) was already included in the appendix which also contained some open ended questions in addition to Likert-scale type questions.    

I suggest the authors to add a final chapter, Limitations and Future research.

Limitation and further research section have been added. It is in line with women involvement in this research as the author believe that there is a need for further in-depth research focusing on the participation of women in order to explore the impact of Merowe dam on women’s contribution to the household economy as well as to the communities.

I wish the authors all the best!

RESPONSE: Thank you very much for your comments. We believe that we are able to satisfy you with our major revision and strongly believe that the present revised version is suitable for publication in the journal.

Reviewer 3 Report

The paper is well-written and well-structured. Despite the limitation of the case study, the research has added value. The choice of research method is adequate.

Author Response

The paper is well-written and well-structured. Despite the limitation of the case study, the research has added value. The choice of research method is adequate.

RESPONSE: Thank you very much for your complementary comments. We have conducted a major revision to accommodate critical comments of other two referees. We strongly believe that the revised version is now suitable for publication in the journal.

Round 2

Reviewer 2 Report

The paper is now much improved.
However, there are still a few concerns regarding this second version of the manuscript.
In Introduction, I suggest the authors to include the  aim and objectives of the paper as well as its originality/ uniqueness.
Chapter 9 should be renamed as  Limitations and future research. In addition, only one limitation is too little for this research. I suggest the authors to identify other limitations for their study.

Author Response

The paper is now much improved.

RESPONSE: Thank you very much for your complimentary comments.

However, there are still a few concerns regarding this second version of the manuscript.
In an Introduction, I suggest the authors include the aim and objectives of the paper as well as its originality/ uniqueness.

RESPONSE: The point has been elaborated and explained in lines 67 – 84. We have addressed this point by adding the overall aim of analysing the social impact of mega-project on the local community. Two objectives were developed (a) identify the range of social issues related to changes the dam brought to local communities, such as displacement, resettlement, access to resources, conflicts, community fabrics, relationships, and health issues for both the displaced and resident peoples; and (b) evaluate the influence of social changes that the dam brought to the region as perceived by communities themselves. Concerning the originality, we have added some of the unique points that drive the research context as well. All revisions are made using track changes.

Chapter 9 should be renamed as Limitations and future research. In addition, only one limitation is too little for this research. I suggest the authors identify other limitations for their study.

RESPONSE: Additional limitation has been added that relate to limited freedom of expression for the participants and researcher must adapt to the government guideline and work around it.  This condition has constrained the research and provides a limited scope of freedom. Therefore, with a new regime in Sudan, further research can be conducted using political ecology study to provide a further understanding of Merowe Dam’s impact on the communities (Section 9 line 981-987).

We strongly believe this round of submission satisfies your concerns and the present version is suitable for publication in the journal.

Thank you once again for your critical insights and comments.